# Amorphous As_2_S_3_ Doped with Transition Metals: An *Ab Initio* Study of Electronic Structure and Magnetic Properties

**DOI:** 10.3390/nano13050896

**Published:** 2023-02-27

**Authors:** Vladimir G. Kuznetsov, Anton A. Gavrikov, Milos Krbal, Vladimir A. Trepakov, Alexander V. Kolobov

**Affiliations:** 1Ioffe Institute, 26 Polytechnicheskaya Str., 194021 St. Petersburg, Russia; 2Institute of Physics, Herzen State Pedagogical University of Russia, 48 Moika Emb., 191186 St. Petersburg, Russia; 3Center of Materials and Nanotechnologies, Faculty of Chemical Technology, University of Pardubice, Nam. Čs. Legii 565, 530 02 Pardubice, Czech Republic

**Keywords:** chalcogenides glasses, As_2_S_3_, transition metal doping, electronic structure and magnetism, density functional theory simulations

## Abstract

Crystalline transition-metal chalcogenides are the focus of solid state research. At the same time, very little is known about amorphous chalcogenides doped with transition metals. To close this gap, we have studied, using first principle simulations, the effect of doping the typical chalcogenide glass As_2_S_3_ with transition metals (Mo, W and V). While the undoped glass is a semiconductor with a density functional theory gap of about 1 eV, doping results in the formation of a finite density of states (semiconductor-to-metal transformation) at the Fermi level accompanied by an appearance of magnetic properties, the magnetic character depending on the nature of the dopant. Whilst the magnetic response is mainly associated with *d*-orbitals of the transition metal dopants, partial densities of spin-up and spin-down states associated with arsenic and sulphur also become slightly asymmetric. Our results demonstrate that chalcogenide glasses doped with transition metals may become a technologically important material.

## 1. Introduction

The discovery of semiconducting properties of chalcogenide glasses by Goryunova and Kolomiets has laid the foundation of a new class of semiconductors, viz. amorphous semiconductors [1]. Due to the presence of lone-pair electrons, i.e., paired *p*-electrons that do not participate in the formation of covalent bonds but occupy the top of the valence band [2], these materials exhibit a number of photo-induced phenomena, such as reversible photostructural changes, photo-induced anisotropy, etc. (for reviews, see [3,4]). In these processes, photo-induced bond modification is caused by bond switching that involves excitation of lone-pair electrons and subsequent formation of a new bonding configuration [5].

On the other hand, transition-metal chalcogenides contain Ch-TM (Ch = chalcogen, TM = transition metal) bonds that are formed by sharing chalcogen lone-pair electrons with empty *d*-orbitals of a transition metal, which can lead to the appearance of hybridized states [6]. This process results in (i) a decrease in the ability of chalcogenide glasses to undergo photostructural changes and should also result in (ii) an appearance of magnetism in doped glasses. It would also be extremely interesting if one could change the local bonding configuration of TM-doped chalcogenide glasses using electronic excitation in a way that would affect its magnetic properties. This would allow one to design glasses with optically switchable magnetic response. As the first step to addressing this issue, we studied, using first principles simulations, the electronic structure and magnetic properties of a prototypical chalcogenide glass a-As_2_S_3_ (amorphous As_2_S_3_) doped with transition metals such as molybdenum, tungsten and vanadium.

Transition metals have the following general electronic configuration: (n−1)d${}^{1÷10}$ns${}^{2}$. As one goes across a row from left to right in the Periodic Table, electrons are generally added to the (n−1)d shell that is filled according to the aufbau principle, which states that electrons fill the lowest available energy levels before filling higher levels, and the Hund rule. The aufbau principle correlates well with an empirical so called (n+l)-rule, which is also known as the Madelung rule [7,8], according to which (i) electrons are assigned to subshells in order of increasing value of n+l, (ii) for subshells with the same value of n+l, electrons are assigned first to the subshell with lower *n*. Later, Klechkovskii proposed a theoretical explanation of the Madelung rule based on the Thomas–Fermi model of the atom [9,10]. Therefore, the (n+l)-rule is also cited as the Klechkovskii rule.

However, the Klechkovskii energy ordering rule applies only to neutral atoms in their ground state and has twenty exceptions (eleven in the *d*-block and nine in the *f*-block), for which this rule predicts an electron configuration that differs from that determined experimentally, although the rule-predicted electron configurations are at least close to the ground state even in those cases. For example, in molybdenum ${}_{42}$Mo, according to the Klechkovskii rule, the 5s subshell (n+l=5+0=5) is occupied before the 4d subshell (n+l=4+2=6). The rule then predicts the electron configuration [Kr] 4d${}^{4}$5s${}^{2}$ where [Kr] denotes the configuration of krypton, the preceding noble gas. However, the measured electron configuration of the molybdenum atom is [Kr] 4d${}^{5}$5s${}^{1}$, despite the fact that the tungsten atom, which is isoelectronic to the molybdenum atom and is an atom of the same group, does have the configuration [Xe] 4f${}^{14}$5d${}^{4}$6s${}^{2}$ predicted by the Klechkovskii rule. By filling the 5d subshell, molybdenum can be in a lower energy state.

The choice of Mo, W and V as transition-metal dopants was determined by the great interest in layered crystalline chalcogenides based on these elements. The interest in transition metal (di)chalcogenides was triggered by the discovery of the fact that in the monolayer limit MoS_2_, which is an indirect-gap semiconductor in the bulk form, becomes a direct-gap semiconductor, which opens up a plethora of possible applications of these materials. The most studied crystalline materials of this class are Mo- and W-dichalcogenides, while vanadium is interesting because its crystalline dichalcogenides possess so called charge-density waves [11,12,13,14,15,16,17,18,19]. At the same time, very little is known about the structure and properties, magnetic in particular, of amorphous chalcogenides containing TMs.

## 2. Simulation Details

The aim of this work was to study theoretically the amorphous magnetism [20,21,22,23] of a prototypical chalcogenide glass a-As_2_S_3_ doped with transition metals (Mo, W and V) by means of computer simulations of electronic and magnetic properties within density functional theory (DFT).

The amorphous phase generation procedure used in this study is a “standard” procedure to obtain in silico a melt-quenched amorphous phase of chalcogenides, which has been used in various publications [24,25]. The idea behind this approach is (i) to randomize the structure at a high temperature, (ii) to equilibrate the structure above the melting point of a material and (iii) to quench the melt in order to obtain the amorphous (glassy) phase.

The amorphous phase was generated with the help of *ab initio* molecular dynamics (AIMD) simulations via the following procedure. As the initial structure we used, the 2×2×3 supercell of the orpiment [26] (a = 22.95 Å; b = 19.15 Å; c = 12.77 Å; α = γ = 90${}^{\circ }$; β = 90.35${}^{\circ }$) consisting of 240 atoms (48 formula units As_2_S_3_) with a density of 3.49 g/cm${}^{3}$. The initial structure was randomized using an NVT-ensemble with a Nosé thermostat by heating the structure to 3000 K during 20 ps followed by cooling to 900 K (a temperature just above the melting point) over a period of 15 ps. The amorphous phase was generated by quenching from 900 K to 300 K over a duration of 15 ps. The times for each stage we used are similar to the times used by others in the field [24,25,27]. A similar procedure was repeated for Mo doped a-As_2_S_3_. A total of 5 Mo atoms were included in the 240 atom Mo-doped cell corresponding to the Mo concentration of 2%.

The generalized gradient approximation (GGA) for the exchange-correlation functional by Perdew–Burke–Ernzerhof (PBE) [28] along with the Projected Augmented Wave (PAW) [29,30] method to describe the electron–ion interaction as implemented in the plane wave pseudopotential code VASP [31,32,33] were used. A value of 260 eV was chosen for plane wave kinetic energy cutoff E_cutoff_. Integration over Brillouin zone (BZ) was accomplished with only Γ-point taking into account the relatively large size of the simulation cell.

To address the issue of the effect of different transition metals on the structure and properties of the glass, we used a simplified approach adopting the following strategy. Since the glass formation process is stochastic and the obtained a-As_2_S_3_:Mo structure is just one of the many possible structures, we replaced Mo species by W and V (assuming that such local structures will statistically be formed), after which the structures were relaxed at 0 K. Full geometry optimization (GOpt) at 0 K of all melt-quenched (mq) amorphous structures, both pure and doped, was accomplished via the Broyden–Fletcher–Goldfarb-Shanno (BFGS) [34,35,36,37] optimizer using spin-polarized DFT within the GGA with the exchange-correlation functional PW91 by Perdew and Wang [38] and the van der Waals (vdW) dispersion correction by Tkatchenko and Scheffler (TS) [39], as implemented in the plane-wave pseudopotential code CASTEP [40,41]. The Vanderbilt ultrasoft pseudopotentials (USPs) [42] were chosen to describe the electron–ion interactions. As: 4s${}^{2}$4p${}^{3}$, S: 3s${}^{2}$3p${}^{4}$; Mo: 4s${}^{2}$4p${}^{6}$4d${}^{5}$5s${}^{1}$; V: 3s${}^{2}$3p${}^{6}$3d${}^{3}$4s${}^{2}$; and W: 5s${}^{2}$5p${}^{6}$5d${}^{4}$6s${}^{2}$; these electrons were assigned to the valence space. Geometry optimization (full relaxation) at 0 K was performed at the Γ-point of BZ with E_cutoff_ = 330 eV. Optimization was carried out until the energy difference per atom, the Hellman–Feynman forces on the atoms and all the stress components did not exceed the values of 5 × 10${}^{-7}$ eV/atom, 1 × 10${}^{-2}$ eV/Å and 2 × 10${}^{-2}$ GPa, respectively. The convergence of the SCF-energy was achieved with a tolerance of 5 × 10${}^{-8}$ eV/atom.

While amorphous structures under consideration are not layered and in this sense are not van der Waals (vdW) solids, the presence of strongly polarizable lone-pair electrons requires the use of vdW corrections in calculations. In the CASTEP code, various types of vdW corrections are implemented for different chemical elements and different exchange-correlation functionals, but not all of them are compatible with all types of GGA-functionals and implemented for all Mo, W and V. The only combination of the GGA functional and the vdW correction implemented in the CASTEP code, available for all three TMs (Mo, W and V), is the combination PW91+TS (Perdew-Wang + Tkatchenko-Scheffler).

Finally, for pure a-As_2_S_3_ and doped a-As_2_S_3_:TM structures, both melt-quenched and melt-quenched with subsequent geometry optimization (mq-GOpt) at 0 K, we calculated different electronic and magnetic properties, including band structure, density of states (DOS), partial and local densities of states (PDOS and LDOS), interatomic distances, atomic, bond and spin Mulliken populations [43,44], total and absolute magnetizations. The Monkhorst–Pack [45] 3×3×5 k-mesh (23 k-points in the irreducible BZ) was used for calculating DOS and PDOS.

The nature of magnetic ordering of a-As_2_S_3_:TM was determined by comparing the calculated values of total and absolute magnetizations. The total magnetization, or in other words the doubled integrated spin density (ISD), gives the doubled total spin of the system. The magnitude of absolute magnetization, or in other words the doubled integrated modulus spin density (IMSD), is a measure of the local unbalanced spin.

Of course, the rather small 240-atom model with just 5 TM atoms may miss some details associated with certain atomic configurations formed in a real glass. Nevertheless, we believe that our model is able to capture the general features of amorphous magnetism in TM-doped chalcogenides glasses.

## 3. Results and Discussion

The mass densities of the simulated equilibrated amorphous structures were verified against the experimental data (Table 1). The presented data demonstrate that they are all slightly lower (3.095–3.321 g/cm${}^{3}$) than that of the crystalline phase, which agrees with the experimental value reported (3.193 g/cm${}^{3}$) [46]. The data presented in Table 1 demonstrate that the discrepancy between the theoretical and experimental a-As_2_S_3_ density values is 3%, which is a very reasonable value. Indeed, its magnitude is comparable to the experimental spread of density data reported by different groups for orpiment (crystalline As_2_S_3_). Obviously for a glass this spread will be even larger.

When doped with TM impurities, the mass densities of amorphous structures increase from 3.095 g/cm${}^{3}$ for undoped a-As_2_S_3_ to 3.159, 3.202 and 3.321 g/cm${}^{3}$ for a-As_2_S_3_ doped with the V, Mo and W impurities, respectively. In this case, we see that the order of increasing mass density upon doping the a-As_2_S_3_ with impurity TM-atoms correlates well with an increase in the atomic number of the impurity TM-atom.

We also note that undoped a-As_2_S_3_ is a semiconductor with a DFT gap of around 1.0 eV as shown in Figure 1(left panel), which is in reasonable agreement with the experimental value of 2.4 eV considering that DFT usually underestimates the gap value by about 50% due to incomplete exclusion of electron self-interaction when using LDA and GGA approximations [47,48].

Figure 2 (left) shows the a-As_2_S_3_:Mo melt-quenched structure. At the right of Figure 2, we show fragments of the structure around the Mo atoms. The following observations can be made. One can see that some of the S atoms are three-fold coordinated, i.e., their lone-pair electrons are consumed to form covalent (donor-acceptor) bonds with Mo. Analysis of the obtained structure shows that of the 25 S atoms that form covalent bonds with Mo species 13 S are three-fold coordinated. In other words, doping with transition metals results in a decreased concentration of lone-pair electrons.

A detailed analysis of the short-range order around molybdenum atoms is difficult as the structure used in the calculations is rather large. That is why to accomplish this analysis fragments of the structure were taken, in which broken bonds were saturated with hydrogen atoms. The differential electron density (charge density difference, CDD), allowing visualization of both covalent bonds (CBs) and lone-pair (LP) electrons, was calculated. CDD is the difference in electron density in the structure under study and the sum of isolated atoms. Consequently the CDD clouds, i.e., an increase in atomic density between atoms, correspond to covalent bonds. Lone pairs are also associated with an increased electron density. An example of electronic distribution visualization is shown in Figure 3. It can be seen that molybdenum atoms are connected with surrounding atoms with covalent bonds. Additionally, lone-pairs formed by *s*-electrons of arsenic atoms are also visible. At the same time, it is interesting to note that CDD corresponding to *p*-lone-pairs of sulfur atoms in some cases are not observed in agreement with the formation of three-fold coordinated sulfur atoms mentioned above.

The effect of Mo-doping on the electronic structure can be seen from Figure 1, where densities of states for the undoped and Mo-doped mq-GOpt structures are given. One can see that spin-up and spin-down DOSs in pure a-As_2_S_3_ are mirror-symmetric in agreement with the experiment (a-As_2_S_3_ is diamagnetic). Here it is important to note that spin-up and spin-down DOSs become different in the doped glass, which is an indication that the material became magnetic. Another apparent difference is the formation of a strong tail at the bottom of the conduction band extending down to the valence band and effectively closing the band gap in the Mo-doped glass.

Molybdenum doping also results in the formation of a finite density of states at the Fermi level, i.e., the material becomes a metal. This is in line with the fact that amorphous MoS_2_ is metallic [49] while the corresponding crystal in its stable 2H form is a semiconductor. The metallic conductivity of amorphous MoS_2_ is due to the formation of a large concentration of metallic Mo-Mo bonds in the amorphous phase. It is remarkable that despite a rather low concentration of Mo atoms in our model, a Mo-Mo dimer is formed, which, again, correlates with the strong tendency of molybdenum to form Mo-Mo bonds in the amorphous phase.

It is not unnatural to assume that isolated Mo atoms and a Mo-Mo dimer contribute differently to the density of states. To address this issue, we removed one of the atoms in the molybdenum dimer, after which the structure was additionally relaxed. In the structure without a dimer, the tail states disappear, while some states in the gap remain. This result allows us to draw a conclusion that the conduction band tail is associated with Mo dimers.

Since Mo atoms all possess different local structures, it is interesting to see which of them (and how) contribute to the magnetic properties. This was done via DFT-method calculating the contributions from individual molybdenum atoms, as well as vanadium and tungsten atoms, to the total spin of the a-As_2_S_3_:TM structures using the well-known Mulliken approximation for the analysis of atomic, bond and spin populations. The results of the Mulliken analysis of spin populations of individual atoms, along with the atomic and bond populations, are presented in Appendix A. The performed DFT calculations showed that due to the different local structure of impurity TM-atoms, not all of them contribute to the total spin of the system. In particular, as can be seen from Table A1 and Table A2, only Mo1 and Mo3 atoms contribute noticeably to the total spin of a-As_2_S_3_:Mo. Similar conclusions can also be drawn for vanadium atoms, where four out of five V atoms make a significant contribution to the total spin for mq a-As_2_S_3_:V (see Table A3) and only two out of five—for mq-GOpt a-As_2_S_3_:V (see Table A4). As regards the antiferromagnetic structure mq a-As_2_S_3_:W with zero total spin, the main noticeable contributions to the total spin come from the projections of spins of opposite signs of atoms W1 and W3 (see Table A5). For equilibrated paramagnetic mq-GOpt a-As_2_S_3_:W structure, all tungsten atoms have zero spins (see Table A6). It should be noted that, in addition to the Mulliken atomic and spin populations, Appendix A also lists the bond populations along with the bond lengths between impurity TM atoms and ligand atoms (see Table A7–Table A12). An analysis of these data allows us to conclude that in the glassy (amorphous) structures a-As_2_S_3_:TM there are mainly two types of hybridization of the orbitals of impurity TM atoms, namely, trigonal-pyramidal (d${}^{1}$sp${}^{3}$) and octahedral (d${}^{2}$sp${}^{3}$). The first hybridization type involves s, p${}_{x}$, p${}_{y}$, p${}_{z}$ and d${}_{z^{2}}$ orbitals in the combination (s+p${}_{x}$+p${}_{y}$)+(p${}_{z}$+d${}_{z^{2}}$). The other one contains another d${}_{x^{2}-y^{2}}$ in addition to the same orbitals in the combination (s+p${}_{x}$+p${}_{y}$+d${}_{x^{2}-y^{2}}$)+(p${}_{z}$+d${}_{z^{2}}$). Of course, in glass hybridization is distorted due to the spread in bond lengths and angles.

While the calculation of the contributions of individual atoms to the total spin of the system using the Mulliken approach is very approximate, the total spin can also be calculated more accurately by integrating the total spin density. Indeed, the CASTEP code enables one to calculate two quantities, which qualitatively determine the type of magnetic ordering, namely the integrated spin density (total spin of the system) and the integrated modulus spin density (a measure of the local unbalanced spin), which allow one to characterize magnetism in materials. Depending on the obtained values, a material can be characterized as being paramagnetic, ferromagnetic, ferrimagnetic or antiferromagnetic. Thus, if both quantities are equal to zero, the material is paramagnetic. Both non-zero quantities and the same magnitude determine ferromagnetic ordering. In the case of non-zero value for the first number and a bigger value for the second, the material is ferrimagnetic. Finally, a zero value for the first number and non-zero for the second characterizes antiferromagnetic ordering of material (the total spin is zero, but the local spin density varies).

In order to verify the appearance of magnetism, we performed spin-up and spin-down PDOS (alpha and beta) calculations. As can be seen from Figure 4, spin-up and spin-down contributions from Mo *d*-electrons are clearly different. At the same time, PDOS of spin-up and spin-down states associated with As and S atoms also become slightly asymmetric which indicates the presence of some contribution to the magnetic properties of the a-As_2_S_3_:TM from As and S atoms when doping with a TM impurity. It is interesting to note that a similar result, namely, an appearance of magnetism on normally diamagnetic S atoms was also observed for edge states of WS_2_ nanosheets [50].

Of significant interest is the observation that the contribution of *d*-electrons substantially changes upon relaxation of the structure. Since the relaxation is done at 0 K when bond breaking is very unlikely to happen and only the existing bond lengths and bond angles slightly change, the result suggests that magnetism in doped a-As_2_S_3_:TM is rather “fragile”. In Figure 5, we compare spin-up and spin-down contributions to PDOS from *d*-electrons of Mo, W and V atoms before and after relaxation. One can see that—as in the case of Mo doping—PDOS shape changes after relaxation. Of special interest is the fact that in the relaxed W-doped a-As_2_S_3_ spin-up and spin-down contributions become identical (paramagnetic state). We believe that the observed fragility of magnetism in doped a-As_2_S_3_ is related to the absence of dark ESR in chalcogenide glasses, which is due to the absence of dangling bonds with unpaired spins caused by the fact that a melt-quenched relaxed glassy structure adjusts to have all bonds saturated [51]. It should be noted that magnetism is also not observed in single crystal transition-metal chalcogenides but appears at sample edges and grain boundaries [6]. Similarly, magnetism in the studied glasses may be associated with the presence of soft modes [52], where some of the bonds may be electron deficient. The latter may be easily affected by slight changes in the local environment.

Analysis of the integrated spin density and integrated modulus spin density shows that the nature of magnetic ordering in a-As_2_S_3_:TM is different for different dopants, namely, the melt-quenched a-As_2_S_3_:Mo and melt-quenched a-As_2_S_3_:V are ferrimagnetic, while the melt-quenched a-As_2_S_3_:W structure is antiferromagnetic. After subsequent relaxation at 0 K of the melt-quenched structure geometry, the Mo- and V-doped structures remain ferrimagnetic, while the W-doped structure becomes paramagnetic. A summary of the results is given in Table 2, which shows the integrated spin densities (total spin of the system) and integrated modulus spin densities (a measure of the local unbalanced spin) for the materials studied. From Table 2, one can see that despite the observed differences in PDOS of as-quenched and 0K-optimized structures the type of magnetism is preserved (ferrimagnetic) in Mo- and V-doped a-As_2_S_3_, while in W-doped a-As_2_S_3_ structural optimization (relaxation) changes the type of magnetic ordering from antiferromagnetic to paramagnetic.

We attribute this unusual behavior of the W-doped amorphous structure to the fact that when the Mo atom is substituted by a bigger W atom, the lattice becomes locally stressed and when this stress is removed during the subsequent optimization of the geometry at 0 K, the nature of the magnetic ordering changes significantly, as a result of which the optimized W-doped a-As_2_S_3_ structure loses magnetic properties. This opens up the interesting possibility of controlling the magnetic properties of the TM-doped amorphous structure by applying external pressure. The study of pressure effect on the magnetic properties of a-As_2_S_3_ is currently underway.

## 4. Conclusions

The electronic configuration of sulphur, s${}^{2}$p${}_{x}^{1}$p${}_{y}^{1}$p${}_{z}^{2}$, means that two of the four *p*-electrons are used to form covalent bonds and two are left as a lone-pair. The accomplished DFT simulations indeed showed that for both structures of a pure glass and a-As_2_S_3_:Mo the majority of As and S atoms satisfy the 8-N rule, i.e., As atoms are three-fold coordinated and S atoms are two-fold coordinated. In the doped glass, some lone-pair electrons are consumed to form Ch-TM bonds.

We demonstrate a strong effect of TM-dopants on the electronic structure of the a-As_2_S_3_ as well as an appearance of a magnetic response. Of special interest is the fact that doping with different transition metals results in the formation of materials with different magnetic properties. Our results suggest that chalcogenide glasses doped with transition metals may become a technologically important material.

## Figures and Tables

**Figure 1 nanomaterials-13-00896-f001:**
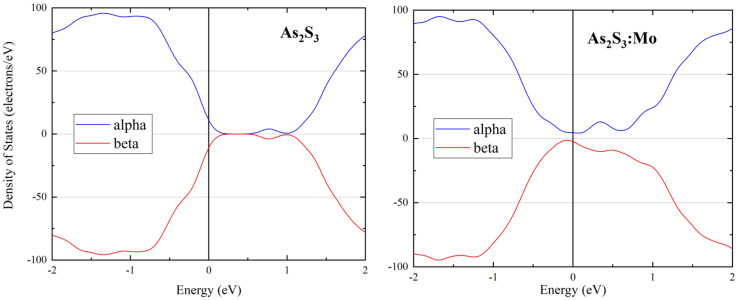
DOS of mq-GOpt structures of a-As_2_S_3_ (**left**) and a-As_2_S_3_:Mo (**right**).

**Figure 2 nanomaterials-13-00896-f002:**
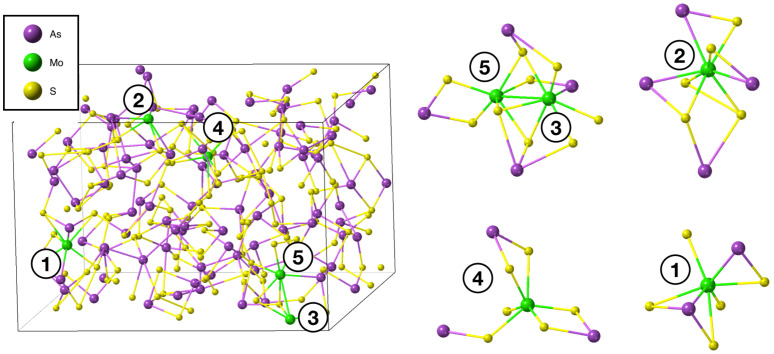
Melt-quenched a-As_2_S_3_:Mo obtained from AIMD simulations (**left**) and fragments of the structure centered around the Mo atoms (**right**). Mo atoms in the a-As_2_S_3_ structure are numbered from 1 to 5 (left panel). In the right panel, local structures around these atoms are shown. As atoms—violet; S atoms—yellow; Mo atoms—green.

**Figure 3 nanomaterials-13-00896-f003:**
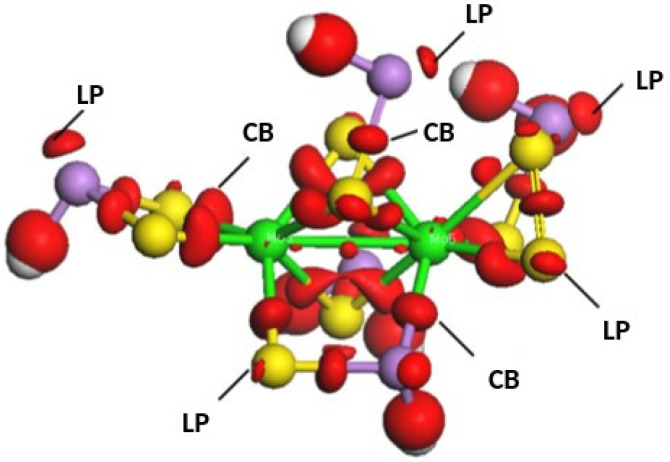
A fragment of Mo-doped a-As_2_S_3_. Dangling bonds are saturated with hydrogen atoms (grey). As atoms—violet; S atoms—yellow; Mo atoms—green. Charge Density Difference is shown in red for covalent bonds (CBs) and lone-pairs (LPs).

**Figure 4 nanomaterials-13-00896-f004:**
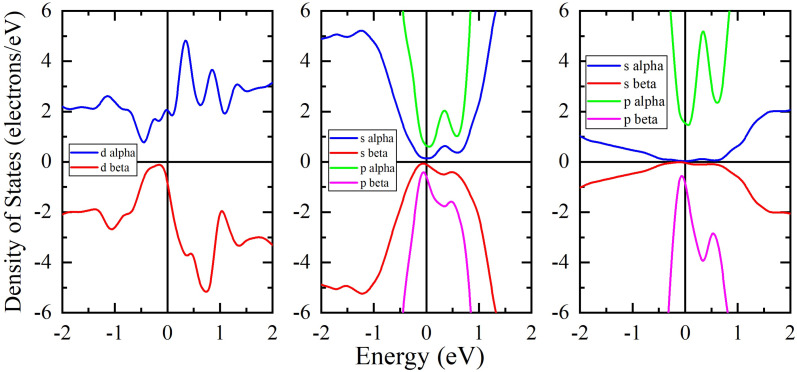
Partial densities of Mo *d*-states (**left**), As sp-states (**center**) and of S sp-states (**right**) for mq-GOpt a-As_2_S_3_:Mo structure.

**Figure 5 nanomaterials-13-00896-f005:**
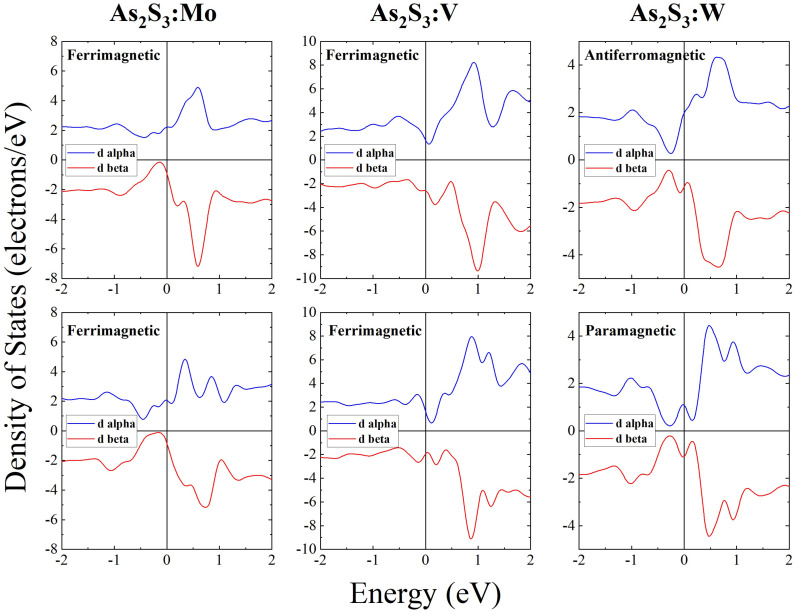
Partial densities of Mo *d*-states, V *d*-states and W *d*-states for mq structures (**upper panel**) and for mq-GOpt structures (**lower panel**).

**Table 1 nanomaterials-13-00896-t001:** Densities of As_2_S_3_ and As_2_S_3_:TM structures.

Type of Structure	Density (g/cm${}^{3}$)
c-As_2_S_3_ (experiment) [26]	3.494
a-As_2_S_3_ (experiment) [46]	3.193
a-As_2_S_3_ mq-GOpt	3.095
a-As_2_S_3_:V mq-GOpt	3.159
a-As_2_S_3_:Mo mq-GOpt	3.202
a-As_2_S_3_:W mq-GOpt	3.321

**Table 2 nanomaterials-13-00896-t002:** Type of magnetic ordering of a-As_2_S_3_:TM structure.

	Melt-Quenched			Melt-Quenched and Relaxed		
**Dopant**	**ISD (** *ℏ* **/2)**	**IMSD (** *ℏ* **/2)**	**Magnetic Ordering**	**ISD (***ℏ***/2**)	**IMSD (** *ℏ* **/2)**	**Magnetic Ordering**
Mo (4s${}^{2}$4p${}^{6}$4d${}^{5}$5s${}^{1}$)	2.00	2.58	ferrimagnetic	2.00	2.54	ferrimagnetic
V (3s${}^{2}$3p${}^{6}$3d${}^{3}$4s${}^{2}$)	2.97	4.36	ferrimagnetic	1.00	1.88	ferrimagnetic
W (5s${}^{2}$5p${}^{6}$5d${}^{4}$6s${}^{2}$)	0.00	1.45	antiferromagnetic	0.00	0.00	paramagnetic

## Data Availability

Additional data can obtained from the authors upon reasonable request.

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
