# Peer review of "Amorphous As2S3 Doped with Transition Metals: An Ab Initio Study of Electronic Structure and Magnetic Properties"

_nanomaterials, 2023, doi:10.3390/nano13050896_

Round 1

Reviewer 1 Report

In this manuscript, the authors studied the doping effect of typical chalcogenide glass As2S3 with transition metals (Mo, W, and V) using first principles simulations. It was found that the magnetic response was mainly associated with d-orbitals of the transition metal dopants, by which the densities of spin-up and spin-down states of As and S became slightly asymmetric. The results are interesting. It is recommended a minor revision before it can be considered for publication in Nanomaterials.

1.      Are there any density effects of doped transition metal on electronic structure and magnetic properties? Please discuss.  

2.      Please explain why study the metals of Mo, W, and V instead of other transition metals.

Author Response

Response to Reviewer 1 Comments

Point 1: Are there any density effects of doped transition metal on electronic structure and magnetic properties? Please discuss.

Response 1: The parameters of the simulated structures should, of course, be verified against the experimental data and density is one such parameter. We added to the revised version of the manuscript data on the density of the equilibrated amorphous phases. As can be seen they are all slightly lower (3.095-3.321 g/cm^3) than that of the crystalline phase, which agrees with the experimental values reported (3.494 g/cm^3) (http://www.webmineral.com/data/Orpiment.shtml#.Y-J8kC8RrOQ). When doped with TM impurities, the densities of amorphous structures increase from its value of 3.095 g/cm^3 for undoped a-As2S3 to values of 3.159, 3.202, and 3.321 g/cm^3 for a-As2S3 doped with the V, Mo, and W impurities, respectively. In this case , we see that the order of increase in density upon doping the a-As2S3 with impurity TM-atoms correlates well with an increase in the atomic number of the impurity TM- atom, which characterizes its size.

Point 2: Please explain why study the metals of Mo, W, and V instead of other transition metals.

Response 2: The interest in transition metal (di)chalcogenides has been triggered by the discovery of the fact that in the monolayer limit MoS2, which is an indirect-gap semiconductor in the bulk form, becomes a direct-gap semiconductor, which opens up a plethora of possible applications of these materials. The most studied crystalline materials of this class are Mo- and W-dichalcogenides. Vanadium, on the other hand, is interesting because its crystalline dichalcogenides possess so called charge-density waves. At the same time, very little is known about the structure and properties, magnetic in particular, of amorphous chalcogenides containing TMs. The choice of Mo, W, and V and transition-metal dopants was determined by the great interest in crystalline chalcogenides based on these elements, as mentioned earlier. We added an appropriate paragraph in the revised manuscript.

Reviewer 2 Report

The authors present a first-principles study of doped As2S3. The subject is moderately interesting. The method employed is up to date. A comparison to experiment is possible for the band gap of undoped As2S3. This comparison yields a deviation of 1.4 eV which cannot be called a 'reasonable agreement'. One might ask the question, if the use of less approximate  pseudopotentials influences this number.
Nevertheless, I believe that publication in Nanomaterials is possible with minor corrections, because the paper is well written, the subject is thoroughly investigated, and the results are well presented.

Minor points:
p. 1: Explain the abbreviation DFT in the abstract.
p. 1: semiconuctor -> semiconductor
p. 1: The abbreviation a-As2S3 is not explained.
p. 6: Figs. 4 and 5 are difficult to read.

Author Response

Response to Reviewer 2 Comments

Point 1: The authors present a first-principles study of doped As2S3. The subject is moderately interesting. The method employed is up to date. A comparison to experiment is possible for the band gap of undoped As2S3. This comparison yields a deviation of 1.4 eV which cannot be called a 'reasonable agreement'. One might ask the question, if the use of less approximate pseudopotentials influences this number.

Response 1: We agree with the referee that a deviation of 1.4 eV of the experimental band gap value from the theoretical one for undoped a-As2S3 is significant. However, the difference (58.3 per cent) fits within the band gap error of 50-60 percents characteristic of the DFT-method. This error is a consequence of incomplete exclusion of electron self-interaction when using LDA and GGA approximations in the DFT-method and has practically nothing to do with the type of pseudopotentials (unless, of course, ghost states are present in them).

If the goal of this work was the accurate calculation of the band gap, then we could achieve it in one of four ways, namely, by using the Tran-Blaha exchange-correlation functional or different hybrid exchange-correlation functionals, by using nonlocal self-interaction correction (SIC) or, finally, using the many-body perturbation theory method (GW-method). However, this work was not aiming at the accurate theoretical calculation of the band gap of undoped a-As2S3, which we mentioned only in passing to complete the picture. Our goal was to study fundamental properties of the doped structures a-As2S3:TM in their ground state, which, unlike the excited states (and the band gap is associated with the excited state), are described with good accuracy by the DFT-method in the GGA approximation, in particular by DFT-method with the exchange-correlation functionals of PBE (Perdew-Burke-Ernzerhof) and its predecessor PW91 (Perdew-Wang). As for the accuracy of the pseudopotentials used, it is well-known that ultrasoft pseudopotentials are the most suitable for the theoretical study of magnetic properties.

Point 2: p. 1: Explain the abbreviation DFT in the abstract.

Response 2: The abbreviation has been spelled out

Point 3: p. 1: semiconuctor -> semiconductor

Response 3: typo corrected.

Point 4: p. 1: The abbreviation a-As2S3 is not explained.

Response 4: a-As2S3 stands for amorphous As2S3. Corrected in the revised text

Point 5: p. 6: Figs. 4 and 5 are difficult to read.

Response 5: The referee is certainly correct when saying that Figs. 4 and 5 are difficult to read. We increased the size of panels in the figures so that they are easier to read.

Reviewer 3 Report

Comments on nanomaterials-2212930:

The current DFT calculation of some As2S3 systems fits the scope of the journal. However, as a computationally orientated work, and the quality of computational regimes and results requires validations and improvements. Below, I provide detailed comments.

The authors have compared the DFT-computed gap (1.0 eV) with experiment (2.4 eV), and a ~1.4 eV deviation is observed. If we consider the percentage/relative deviation with the computational result 1.0 eV, this deviation is as large as 140%. To avoid this huge relative error, the authors report, in the first paragraph of section 3, that the relative deviation with respect to experiment is 50%.

Concerning the selection of the Hamiltonian (electronic structure calculation), I understand that PBE is commonly applied in this field and serves as a robust option, but in many cases it is not the correct or the best selection. Further there are many revised versions with varying performances in specific cases. The authors should benchmark the accuracy levels of different DFT levels and then pick one with balanced accuracy and cost. The current calculation simply selecting a DFT level without justification is rather dangerous. Further, as still obvious discrepancy exists between computational and experimental results, it is difficult to validate the reliability of the other observations based on such calculations.

Aside from the Hamiltonian, many other settings are applied without justification. For example, the simulated annealing procedure of 20 ps heating to 3000 K, 15 ps cooling to 900 K and then 15 ps further to 300 K is simply applied and the authors state that the amorphous phase is well reached, which seems rather casual and lacks of detailed validation. It seems unclear how this 20 ps -> 15 ps -> 15 ps setting to be sufficient? Further, there should be some criterion assessing the quality of the generated structure. Otherwise, any calculations with shorter (more aggressive) protocols (e.g., 1 ps -> 1 ps -> 1 ps) could also produce reasonable estimates.

Does the densities of the simulated systems agree with experimental values? This is the most straightforward comparison for a given system, although accurate reproduction of mass density does not really guarantee the correctness of the other observables (e.g., diffusion).

I wonder how many structures (after simulated annealing) are included in the final calculations producing the figures and tables. Does different structures lead to different results? What is the magnitude of fluctuations?

Author Response

Response to Reviewer 3 Comments

Point 1: The authors have compared the DFT-computed gap (1.0 eV) with experiment (2.4 eV), and a ~1.4 eV deviation is observed. If we consider the percentage/relative deviation with the computational result 1.0 eV, this deviation is as large as 140%. To avoid this huge relative error, the authors report, in the first paragraph of section 3, that the relative deviation with respect to experiment is 50%.

Response 1: We agree with the referee that a deviation of 1.4 eV of the experimental band gap value from the theoretical one for undoped a-As2S3 is significant. However, this difference (1.4/2.4=0.58, i.e. 58% smaller than the real value) fits within the usual band gap error characteristic of the DFT-method. This error is a consequence of the incomplete exclusion of the electron self-interaction when using the LDA and GGA approximations in DFT-method.

Point 2: Concerning the selection of the Hamiltonian (electronic structure calculation), I understand that PBE is commonly applied in this field and serves as a robust option, but in many cases it is not the correct or the best selection. Further there are many revised versions with varying performances in specific cases. The authors should benchmark the accuracy levels of different DFT levels and then pick one with balanced accuracy and cost. The current calculation simply selecting a DFT level without justification is rather dangerous. Further, as still obvious discrepancy exists between computational and experimental results, it is difficult to validate the reliability of the other observations based on such calculations.

Response 2: In this work, we studied the effect of impurities of Mo, W, and V on the appearance of magnetism in an amorphous structure of a-As2S3 doped with them. The choice of Mo, W, and V as transition-metal dopants was determined by the great interest in layered crystalline chalcogenides based on these elements. While amorphous structures under consideration are not layered and in this sense are not van der Waals (vdW) solids, the presence of strongly polarizable lone-pair electrons requires the use of vdW corrections in the calculations.

In the CASTEP code various types of vdW corrections are implemented for different chemical elements and different exchange-correlation functionals, but not all of them are compatible with all types of GGA-functionals and implemented for all Mo, W, and V. The only combination of the GGA functional and the vdW correction implemented in the CASTEP code, available for all three TMs (Mo, W, and V) is the combination PW91+TS (Perdew-Wang + Tkatchenko-Scheffler). This level of DFT simulations is typically used to study the amorphous phase of chalcogenides [Elliott, Micoulaut, M., Piarristeguy, A., Flores-Ruiz, H. and Pradel, A., 2017. Towards accurate models for amorphous GeTe: Crucial effect of dispersive van der Waals corrections on the structural properties involved in the phase-change mechanism. Physical Review B, 96(18), p.184204]

The discrepancy between computational and experimental results noted by the referee concerns only the band gap (which is an excited state property) for undoped a-As2S3 and is explained by the existence of the well-known «band gap problem in the DFT», due to incomplete exclusion of the electron self-interaction when using the LDA and GGA approximations in DFT-method.

Point 3: Aside from the Hamiltonian, many other settings are applied without justification. For example, the simulated annealing procedure of 20 ps heating to 3000 K, 15 ps cooling to 900 K and then 15 ps further to 300 K is simply applied and the authors state that the amorphous phase is well reached, which seems rather casual and lacks of detailed validation. It seems unclear how this 20 ps -> 15 ps -> 15 ps setting to be sufficient? Further, there should be some criterion assessing the quality of the generated structure. Otherwise, any calculations with shorter (more aggressive) protocols (e.g., 1 ps -> 1 ps -> 1 ps) could also produce reasonable estimates.

Response 3: The procedure used in our work is a “standard” procedure to obtain in-silico melt-quenched amorphous phase of chalcogenides and has been used in various publications [Konstantinou, K.; Mavračić, J.; Mocanu, F.C.; Elliott, S.R. Simulation of Phase-Change-Memory and Thermoelectric Materials using Machine-Learned Interatomic Potentials: Sb2Te3. physica status solidi (b) 2020, 258, 2000416. https: //doi.org/10.1002/pssb.202000416.

Mocanu, F.C.; Konstantinou, K.; Mavračić, J.; Elliott, S.R. On the Chemical Bonding of Amorphous Sb2Te3. physica status solidi (RRL) – Rapid Research Letters 2020, 15, 2000485. https://doi.org/10.1002/pssr.202000485.].

The idea behind this approach is (i) to randomise the structure at a high temperature, (ii) to equilibrate the structure above the melting point of a material and (iii) to quench the melt in order to obtain the amorphous (glassy) phase. The times for each stage we used are similar to the times used by others in the field. We modified the manuscript to reflect this issue.

Point 4: Does the densities of the simulated systems agree with experimental values? This is the most straightforward comparison for a given system, although accurate reproduction of mass density does not really guarantee the correctness of the other observables (e.g., diffusion).

Response 4: The referee is certainly correct when saying that the parameters of the simulated structures should be verified against the experimental data and density is one such parameter. We added to the revised version of the manuscript data on the density of the equilibrated amorphous phases. As can be seen they are all slightly lower ( 3.095-3.321 g/cm^3) that of the crystalline phase, which agrees with the experimental values reported (3.494 g/cm^3) (http://www.webmineral.com/data/Orpiment.shtml#.Y-J8kC8RrOQ)

Point 5: I wonder how many structures (after simulated annealing) are included in the final calculations producing the figures and tables. Does different structures lead to different results? What is the magnitude of fluctuations?

Response 5: The referee is correct when raising the issue of how well our 240-atom model reproduces a real amorphous solid, where the number of combinations of possible local structures is essentially unlimited. It is clear that even a very large model of 500 atoms (considering the possibility of ab-initio molecular dynamics (AIMD) method simulations) would not be much more reliable than our modestly large model of 240 atoms. Consequently, any AIMD-method simulation will only be an attempt to predict the properties of an amorphous solid by analysing a limited number of local atomic configurations. In our work we demonstrate that TM atoms tend to form dimers, which is in agreement with the observed large concentration of TM-TM bonds in amorphous transition-metal dichalcogenides despite their absence in the stable 2H crystalline phase. Hence it is very reasonable. As regards magnetic properties, our results show that the energy differences among different magnetic configurations are rather small and, as a consequence, the amorphous magnetism of doped As2S3 glass is fragile. We observed these results for a system containing 5 TM atoms. If, as rightfully suggested by the referee, we repeated the procedure for several different structures, we would have observed similar results for 10, 15, 20 etc atoms, which does not guarantee that in a real glass with millions of atoms some would not behave differently. We believe that, since we are trying to capture the most fundamental properties of doping a chalcogenide glass with TMs, and not all the details, the use of a modestly large model of 240 atoms is justified.

Round 2

Reviewer 3 Report

Comments on nanomaterials-2212930.R1:

It’s good to see that some face-to-face comparison with experiment has been added to the paper. Concerning the density reported in Table 1, there should be uncertainties estimates for this statistical observable. The authors are investigating a simulation box (ensemble) and a proper average-over-snapshots estimate is computed. For such statistical observables, there should be proper estimates of its statistical uncertainty to reflect the fluctuations inside the system. The same applies to all other properties extracted from the simulation trajectory.

The authors state that their simulated annealing procedure is a ‘standard’ approach applied in many cases. However, this does not guarantee that the method produces satisfactory results in their specific case. This problem exists in all application studies of computational tools, and some validation tests of the simulation outcome as a sanity check must be provided for solidity. Concerning the current system, I believe the density profile along some axis (e.g., z-axis) to be a useful property to check. I.e., the authors should plot the density profile along e.g., the z-axis to validate whether the density profile exhibits the expected behavior of the amorphous phase.

Concerning the absolute value of density estimates presented in Table 1, if I understand it correctly, the only face-to-face comparison between experiment and computation is the a-As2S3 system. The simulation outcome is 3.095 g/cm3, while the experimental value is 3.193 g/cm3. The current discussion presented above Table 1 seems to compare a-As2S3 with and without impurities with experiment, which needs clarifications. Further, the calculation error ~0.1 g/cm3 is not a small value, although it is not very large considering the percentage error.

Round 3

Reviewer 3 Report

Comments on nanomaterials-2212930.R2:

There are two issues in the current version.

First, the uncertainty estimates should be provided in Table 1, not just presented in the response letter. The authors are computing the mass density with a simulation trajectory. This is a statistical observable, and fluctuations between snapshots are expected.

Second, I don't know why the authors refuse to provide a check of the amorphous phase of their system. Applying a sampling protocol with successes in many cases does not guarantee its success in newly encountered situations. For example, we run for 100 ps and the density of water could be converged, but for high-viscosity liquids the sampling time should be extended to e.g., 10 ns. Concerning the specific case under study in the current paper, the authors could argue that the density profile proposed by the reviewer is not a proper selection, but they should come up with some other useful and valid checks themselves. Simply responding against reviewer's comments does not improve the solidity of their work. It is widely acknowledged that in scientific studies, we should support solidity with solid numerical data (statistics or figures), rather than stating successes in many other publications. 
